# Role of Bronchial Artery Embolization as Early Treatment Option in Stable Cystic Fibrosis Patients with Sub-Massive Hemoptysis: Personal Experience and Literature Review

**DOI:** 10.3390/jcm11216432

**Published:** 2022-10-30

**Authors:** Chiara Floridi, Pietro Boscarato, Claudio Ventura, Alessandra Bruno, Nicolo’ Rossini, Michela Baldassari, Cecilia Lanza, Benedetta Fabrizzi, Roberto Candelari, Andrea Giovagnoni

**Affiliations:** 1Department of Clinical, Special and Dental Sciences, University Politecnica delle Marche, 60126 Ancona, Italy; 2Division of Special and Pediatric Radiology, Department of Radiology, University Hospital “Umberto I—Lancisi—Salesi”, 60126 Ancona, Italy; 3Division of Interventional Radiology, Department of Radiological Sciences, University Politecnica delle Marche, 60126 Ancona, Italy; 4Cystic Fibrosis, Regional Reference Center, Child Department, United Hospital, 60126 Ancona, Italy

**Keywords:** hemoptysis, cystic fibrosis, bronchial artery embolization (BAE), chest angio-CT

## Abstract

(1) Background: We describe our experience with cystic fibrosis (CF) patients treated with bronchial artery embolization (BAE) for sub-massive hemoptysis to understand if early treatment of sub-massive hemoptysis can reduce the volume of any subsequent bleedings. (2) Materials: We performed a retrospective study including CF patients who underwent angiographic procedures for BAE following sub-massive hemoptysis, from March 2016 to December 2021. All patients underwent an initial chest angio-CT study. BAE was realized with microspheres or coils. (3) Results: Thirteen patients were included, subjected to at least one BAE after sub-massive hemoptysis, for a total of 19 procedures. Technical success was 94.7%; in a single case, the catheterization of the bronchial arterial feeder was not achievable and the procedure was repeated. Primary clinical success was 92.3%; secondary clinical success was 69.2%. Relative clinical success was 85%. A higher incidence of recurrent hemoptysis following treatment with coils was observed (100% of cases) compared to treatment with microspheres (54.5% of cases) χ^2^ = 5.43 (*p* < 0.05). (4) Conclusions: BAE is a safe and effective method for the treatment of hemoptysis in CF patients; it should be practiced not only after massive or recurrent hemoptysis but also in patients with sub-massive bleeding to improve their life expectancy and quality of life.

## 1. Introduction

Cystic fibrosis (CF) is a congenital recessive autosomal disease caused by the transmission of a mutation in the CFTR (Cystic Fibrosis Transmembrane Conductance Regulator) gene. Hemoptysis, which is blood emission, coming from high airways, with cough, is a frequent event in CF patients and occurs in almost 10% of them. Hemoptysis quantification is difficult and it is not always useful in clinical practice despite its relevance in treatment planning. It is usually classified as mild, moderate and massive. In the literature, there are different quantitative limits to define massive hemoptysis. It is usually identified as a bleeding with volume >240 mL in only one episode, or as repetitive bleedings each with volume >100 mL for days or weeks [1,2]. In the literature, there is no agreement on the definition of sub-massive hemoptysis. In CF patients, hemoptysis is often a minor and self-limiting event. Massive hemoptysis is less common (4–6% of cases) than sub-massive [2,3,4,5], but when it occurs, it represents a serious complication given that it is related to higher morbidity, a higher hospitalization rate, increased rapidity in pulmonary function loss and higher mortality (5.8–16.1%) [1,6,7,8]. Approximately 4.1% of CF patients will have at least one episode of massive hemoptysis during their life. This rate will increase in the future [1,5,7,9,10,11,12]. In CF patients, hemoptysis is caused by newly formed arterial vessels originating from bronchial arteries. Their development is the result of the neoangiogenesis process induced by inflammation and hypoxia related to CF. High arterial pressure into these vessels causes their rupture inside airways and subsequent hemoptysis. In 5% of cases, pulmonary arteries can be the origin of hemoptysis, in relation to direct damage of these arteries (tuberculosis, cancer, pulmonary abscess, pulmonary trauma, arteriovenous malformation, aorto-pulmonary fistula) [1,2,3,11,12,13,14]. Massive hemoptysis can be present both in patients with severe and non-severe pulmonary CF; in the second case, it is related to a specific localized inflammation area which leads to hemoptysis.

In general, hemoptysis must be quickly treated in hemodynamic or respiratory unstable patients. Bronchoscopy is usually the first choice in these cases to achieve a stable condition and to isolate airways. In hemodynamic stable patients, instead, the first target is identifying the site of the bleeding to permit selective bronchial artery embolization (BAE), which is the standard of care treatment in life-threatening hemoptysis.

Chest X-ray is the first useful exam used to identify the site of the bleeding in terms of right or left lung. Chest angio-CT is the better exam in identifying the site and the cause of the bleeding and in recognizing the anatomical course of the main and collateral bronchial arteries and non-bronchial arteries. The following angiographic study, before selective catheterization, reduces the risk of severe complications given that it allows a final anatomical evaluation. In 10% of cases, active arterial bleeding is visible in angiography, whereas in other cases, the site of bleeding can be supposed in relation to some arterial vessels’ characteristics that suggest a possible bleeding origin, such as vascular hypertrophy (diameter > 3 mm), tortuous/dysmorphic vessels, hypervascularized areas in the lung parenchyma, aneurysms or shunts (between bronchial artery and pulmonary vein or between bronchial artery and pulmonary artery).

Some authors support that BAE is indicated only for the treatment of massive hemoptysis and of chronic recurrent hemoptysis in hemodynamic stable CF patients [15], and there is no consensus on the usage of BAE for treating sub-massive hemoptysis in hemodynamic stable patients, who are still usually treated with conservative therapy. Despite this, an early BAE—that is, a bronchial artery embolization performed when the bleeding is still sub-massive—appears to have some relevant advantages. In fact, massive hemoptysis is the sign of a severe alteration of bronchial artery walls and it is a dangerous event for the patient’s life, with negative prognosis and significant reduction in the quality of life itself [4]. Moreover, the early control of the bleeding through BAE may help to improve the patient compliance with medical therapy by reducing the number of bleeding episodes, the quantitative of blood sputum and the medical therapy breaks and improving patient stability for lung transplantation [3,10]. In the past 30 years, BAE has been a safe and effective treatment to control hemoptysis in CF patients. Its success rate is from 70% to 99% and the complication rate is 0–6.6% [16,17,18,19,20,21]. Bleeding recurrences after BAE treatment are frequent events in the literature, but there is no one study that evaluates BAE’s impact on the entity of recurrences (that is, on the bleeding volumes of recurrences) and in particular that evaluates the entity of recurrences in sub-massive hemoptysis treated with BAE.

The aim of our study was to report the experience of our institute with adult CF patients treated with BAE for sub-massive hemoptysis to evaluate technical success, primary/secondary clinical success and the entity of recurrences, when these were present (relative clinical success), in order to understand if early treatment of sub-massive hemoptysis can reduce the volume of subsequent bleedings, reducing at the same time their dangerousness.

## 2. Materials and Methods

### 2.1. Patients

We led a retrospective study from March 2016 to December 2021 at Ospedali Riuniti of Ancona. The ethics committee of our hospital approved this study.

Patients aged 21 to 52 years with CF who underwent at least one angiographic procedure for embolization of the bronchial arteries following sub-massive hemoptysis (defined as loss >80 mL in 24 h but <240 mL in 24 h) were included.

All cases that did not have angio-CT before treatment or for which images were not available were excluded from this study.

Patients < 18 years of age were excluded.

Patients who had already undergone treatment in other centers and those who refused follow-up at our institution were excluded.

### 2.2. Procedural Planning

Before the angiographic study, all patients underwent an initial chest angio-CT study, performed with acquisition without contrast, angiographic acquisition aimed at the study of pulmonary arterial vessels after bolus injection of contrast, and monitoring with bolus tracking technique in forced inspiration and a large field of view.

The acquisitions were performed using a low dose protocol both in 16-layer CT equipment (the only one available for examinations performed until 2018) and using the technology of the third-generation dual-source Dual Energy CT available to our department since 2018 (Somatom Force, Siemens Healthineers, Forchheim, Germany).

The angio-CT study was aimed at evaluating the bronchial arteries, their morphology and the possible presence of non-bronchial arterial collaterals; in particular, specific attention was paid to the treatment of the caliber of the bronchial arteries, both those subjected to subsequent treatment and those not treated (Figure 1).

For the purpose of choosing the site to be treated, a difference in caliber of >2 mm was considered significant.

When the difference in caliber between the two sides was not significant, the treatment was still performed when angiographic parameters were present, such as altered morphology of the bronchial arterial vessel and tortuosity, as well as the presence of a scarce reduction in caliber along its course up to the hilum and the degree of intra-parenchymal, more peripheral vascular changes not noticeable on CT angiography.

### 2.3. BAE Procedure

All patients were given an explanation of the procedures, including benefits and risks, and informed consent was always acquired. All the procedures were performed with local anesthesia in the puncture site. Femoral access was always used. A catheter was first introduced into the common femoral artery through the Seldinger technique in order to explore the aorta with an aortography. In this way, the site of bleeding, when present, or the anomalous vessels, in terms of hypertrophy (diameter > 3 mm) or in terms of tortuous and dysmorphic course, were identified. A micro-catheter was subsequently introduced into the main catheter in order to perform a selective bronchial arteriography. The micro-catheter has minimal traumatism in the vessel wall. Then, BAE was realized through the release of microsphere or coils (Figure 2).

The choice of the material used for BAE was based on its characteristics, particularly on ease of release, occlusion and recanalization duration, and the dimensions of the material.

The materials used in this study were microspheres (Embozene 500 and 700 μm) and coils. Particles with diameter <300 μm are not recommended for BAE because the anastomosis between bronchial arteries and pulmonary arteries have a diameter of approximately 325 μm [22], and for this reason, an embolizing material with a diameter inferior to the one of the anastomosis can lead to a shunt effect and a consequent pulmonary micro-embolism or pulmonary infarct. Gelatin sponge was not used, given that it is not effective [23]. Coils are useful to permit the proximal occlusion of the treated vessel but are also associated with collateral vessel development. Moreover, the spiral proximal positioning prevents future endovascular approaches blocking the treated vessel. These are associated with a higher recurrence rate [23].

### 2.4. Endpoints

A series of outcomes were considered:(a)The safety of the procedure—that is, the rate of peri- and post-procedural complications and the mortality related to the procedure; CIRSE classification [24].(b)Technical success, defined as the percentage of procedures in which effective exclusion of the vessel to be treated was achieved with immediate post-procedure hemoptysis control; both the inability to catheterize the bronchial artery and all cases in which the final angiographic study is considered unsatisfactory were considered technical failure.(c)Primary clinical success, defined as the absence of recurrence of hemoptysis and hemoptysis-related mortality 1 month after the procedure.(d)Secondary clinical success, defined as the absence of recurrence of hemoptysis and hemoptysis-related mortality 6 month after the procedure.(e)Relative clinical success, defined as a reduction in blood quantitative at the first hemoptysis episode after BAE compared with blood quantitative at the hemoptysis episode before BAE.(f)Secondary outcomes were considered: the influence of the embolic agent and Pseudomonas infection rate in relapsing episodes and pre/post-procedural FEV1.

### 2.5. Statistical Analysis

The data were collected in a descriptive analysis and analyzed using GrahPad Prism 9.1.1 (San Diego, CA, USA), using the chi-square test for the qualitative variables; for the qualitative variables and for the quantitative variables with abnormal distribution, the Wilcoxon test was used.

For the quantitative variables, Student’s t and Spearman’s correlation analysis were used.

The significance levels were set at 5%.

The analysis of time free from illness was performed using the Kaplan–Meier test.

## 3. Results

Thirteen patients (seven male, six female) with a mean age of 30 years (±9.5) were included, affected by cystic fibrosis with clinical–anthropometric characteristics summarized in Table 1, subjected to at least one embolization procedure of the bronchial arteries for the onset of sub-massive hemoptysis, for a total of 19 procedures (Table 2).

Revision of CT angiography did not show ongoing arterial bleeding in any patient; all patients had at least one bronchial artery ectasia (Table 3).

Complications were observed in four procedures (21%): three cases (15.8%) reported chest pain and one case (5.2%) reported femoral access site pain (Grade 1–2 CIRSE).

No major complications were recorded in the post-treatment clinical evaluation (Grade 3–6 CIRSE).

Technical success of 94.7% was achieved; in a single case, the catheterization of the bronchial arterial feeder was not achievable because of vasospasm, and the procedure was repeated after six days.

Primary clinical success was 92.3%. Secondary clinical success was 69.2% (Figure 3).

The relative clinical success was 85%; only in two cases was the amount of blood at the first episode of hemoptysis after BAE higher than before treatment (*p* = 0.0466) (Figure 4).

A higher incidence of recurrent hemoptysis following treatment with coils was observed (100% of cases) compared to treatment with microspheres (54.5% of cases), χ^2^ = 5.43 (*p* < 0.05).

No further statistically significant results were obtained from the data analysis; there were no statistically significant differences between the bleeding hemoptysis recurrence rate and Pseudomonas infection. 

FEV1% (relative to predicted for age and sex) assessed before (52.7 ± 14.9) and after the BAE procedure (53.1 ± 16.9) was not significantly modified.

## 4. Discussion

Our experience suggests that BAE is a safe and effective procedure for the treatment of sub-massive hemoptysis in CF adult patients, and in particular, that early BAE is significantly associated with recurrences characterized by equal or lower volumes of blood sputum than pre-procedural hemoptysis volumes. For this reason, we strongly believe that BAE must be used for the treatment of hemoptysis also in sub-massive episodes in order to reduce the recurrence rate and recurrence severity in terms of the amount of blood, and consequently to improve patients’ quality of life and compliance with bridging therapy before transplantation. This statement is also supported by the fact that FC patients have a high tendency to have recurrences (from 30% to 60%) [3,5], and for this reason, an early BAE treatment reduces the number, volume and thus the severity of these episodes.

Moreover, we confirm that the materials used for embolization treatment influence the recurrence rate of hemoptysis; in particular, coils are associated with a higher recurrence rate than microspheres [23]. Coil usage is linked to two main problems: first, the high rate of recanalization, and second, the preclusion of peripheral access in case of recanalization through collateral vessels [11]. Few studies support that coil treatment is safe and effective for hemoptysis [25]. For this reason, we strongly believe that microspheres must be used for BAE in sub-massive hemoptysis treatment.

Up to now in the literature, different studies have described BAE as a successful therapeutic procedure in terms of safety and effectiveness for the control of massive or recurrent hemoptysis in CF patients [2,4,6,15] (Table 3), and in the past, before the development of this procedure, patients were treated conservatively or surgically [1,26,27,28,29]. In only two studies [3,10], BAE was considered for sub-massive hemoptysis. To our knowledge, our retrospective cohort study is the largest in the literature evaluating BAE for sub-massive hemoptysis in CF adult patients.

Indeed, Antonelli et al. [3] evaluated the efficacy of BAE to treat sub-massive hemoptysis in a small CF patient cohort. They showed a technical success of BAE of 100%, and comparing the BAE group with the conservative group, they showed a lower frequency of recurrent bleeding and pulmonary exacerbations and complications without lung functional test modifications. In accordance with that study, we confirmed the results of BAE in a larger group of CF patients with sub-massive hemoptysis.

Differently, Flight et al. [10] evaluated the role of BAE, comparing a group of CF patients with massive hemoptysis to a second one with sub-massive hemoptysis. They did not find any differences between the two groups in terms of recurrences, concluding that the role of BAE in sub-massive hemoptysis is still unclear.

Nevertheless, none considered the severity of recurrences after BAE in terms of the quantity of blood. This parameter was considered in our study, adding another dimension to corroborate the importance of early BAE.

Sweezey et al. [4] showed that cessation of hemoptysis is usually achieved more rapidly with BAE than with medical therapy.

We demonstrate that early BAE has a success rate with values comparable to those of treatment of massive hemoptysis (summarized in Table 3).

In our experience, the most frequent complication was chest pain development after the procedure. Pain was not significant, unlike in other investigations that reported severe and long-lasting pain needing adequate analgesia [6,15], and no dysphagia or post-embolization fever were present in our experience, unlike in other studies in which these minor complications occurred. These data underline the safety of early BAE.

Moreover, in the literature, Pseudomonas infection seems to be related to a more rapid decline in lung function but does not appear to have a clear association with massive hemoptysis [7,11]. Our study shows that it does not appear to be associated with bleeding recurrence rate, whereas Silveira et al. stated that this kind of infection is related to longer hemoptysis-free periods, with a sort of protective effect [12]. In accordance with Silveira et al., we support that the role of Pseudomonas infection in FC patients with hemoptysis is still unclear.

Finally, although one study [5] shows that patients with hemoptysis have an accelerated decrease in FEV1 after BAE, our study did not show a significant difference in FEV1 before and after BAE. This result confirms that the early BAE procedure is safe and does not influence lung functional parameters.

The present study has several limitations. First, it was conducted in a single medical center with a relatively small sample size, limiting its statistical power.

In summary, our results suggest that BAE should be practiced not only in CF patients with massive or recurrent hemoptysis but also in those who have sub-massive episodes of bleeding in order to improve their life expectancy and to ensure a better quality of life.

## Figures and Tables

**Figure 1 jcm-11-06432-f001:**
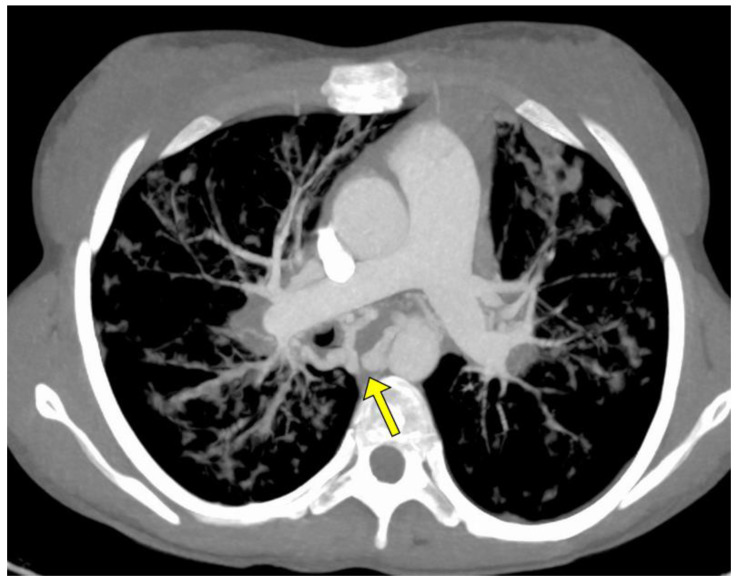
MIP axial Angio-CT showing ectatic and tourtuos (arrow) bronchial arteries in a young woman with sub-massive hemoptysis in CF.

**Figure 2 jcm-11-06432-f002:**
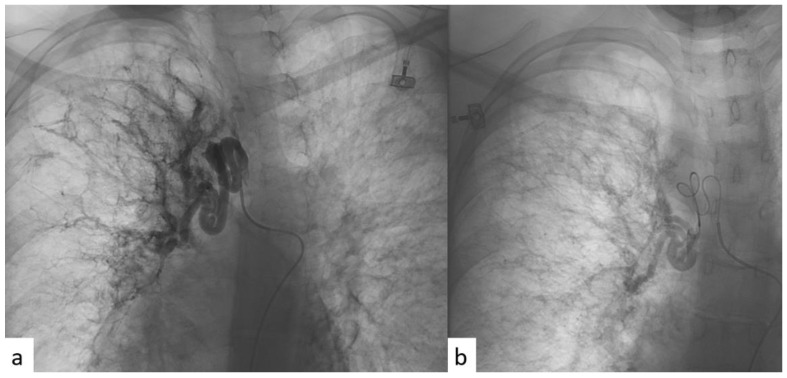
(**a**) Angiographic imaging pre-procedure, performed through selective catheterism of right bronchial artery, showing ectatic and tourtuous bronchial arteries coming from upper and medium lung lobes. (**b**) Angiographic imaging post-procedure, after super-selective BAE with microspheres, showing reduction in blood intake at the level of the treatment.

**Figure 3 jcm-11-06432-f003:**
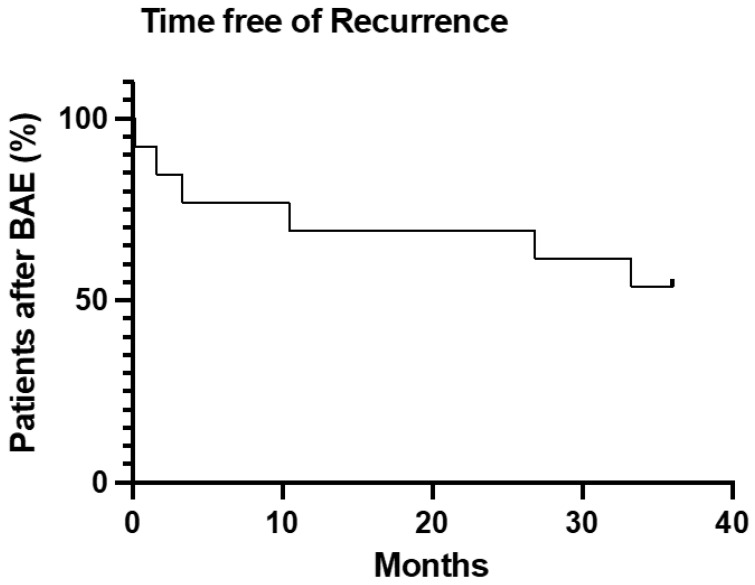
Survival analysis of time free from hemoptysis or death after BAE treatment.

**Figure 4 jcm-11-06432-f004:**
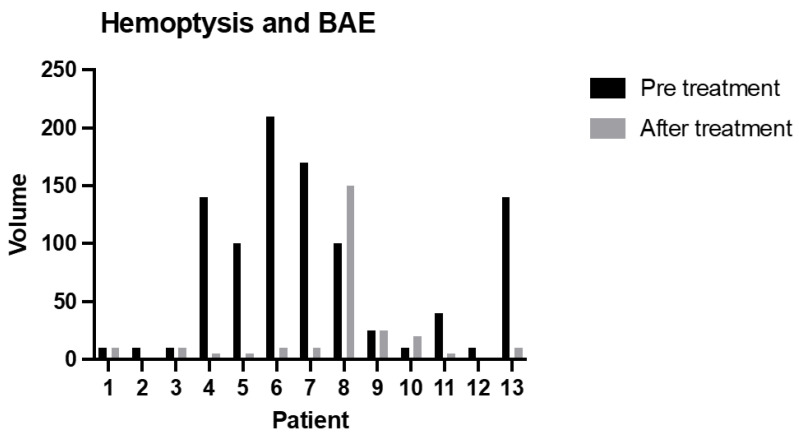
The amount of blood sputum (mL) before treatment and during the relapse episode (after treatment) (*p* = 0.0466).

**Table 1 jcm-11-06432-t001:** Characteristics of the patients at the first hemoptysis episode requiring bronchial artery embolization (n = 13).

Patient Characteristics	Patients	
Sex		
Male	7	53.8%
Female	6	46.2%
Age, Years	30	±9.5
Age at Diagnosis	3	(14)
**Ethnicity**		
White	13	100%
**Respiratory Infection ***		
Stenotrophomonas Maltophilia	2	15.4%
MRSA	3	23.1%
Alcaligenes xylosoxidans	1	7.7%
Mycobacterium Tuberculosis	1	7.7%
Acinetobacter Baumannii	1	7.7%
Pseudomonas Aeruginosa	8	53.8%
**Pulmonary function**		
FEV1% predicted pre procedure	52.7	±14.9

Qualitative data are expressed as absolute (n) and relative (%) frequencies. Quantitative data are expressed as mean ± SD or median and [IQR]. * Infection at the first embolization episode; some patients are co-infected.

**Table 2 jcm-11-06432-t002:** Characteristics of the procedures of embolization (n = 19).

Procedures	ID Pz.	Vessel Type Treated with BAE	Caliber of Treated Vessel (mm) Pre-Procedure CT	Untreated Vessel Size (mm) Pre-Procedure CT	∆ Significant Caliber between Left and Right?	Arterial Vessel Studied in Diagnostic Angiography
1	1	Left	5	2	Yes	Bronchial left
2	1	Left	5	3	Yes	Bronchial left
3	2	Right	6	2	Yes	Bronchial right
4	2	Left + Right	7.5	3.5	Yes	Bronchial right and left *
5	3	Right	6	3.5	Yes	Bronchial right
6	4	Right	7	6	No	Bronchial right and left
7	5	Right	3.5	3.5	No	Bronchial right and left
8	6	Right	8	5	Yes	Bronchial right
9	7	Right	5	2	Yes	Bronchial right
10	7	Right	5	2	Yes	Bronchial right and left
11	7	Right	5	2	Yes	Bronchial right
12	8	Right	3	2	No	Bronchial right and left
13	8	Right	4	2	No	Bronchial right and left *
14	9	Left	8	7	No	Bronchial left *
15	10	Right	7	1	Yes	Bronchial right
16	10	Right	7	1	Yes	Bronchial right
17	11	Right	5	3	No	Bronchial right and left
18	12	Left	7	4.5	Yes	Mammary left
19	13	Right	3.5	0.5	Yes	Bronchial right

* Patients for whom bilateral angiography was performed despite the significant difference between the two sides.

**Table 3 jcm-11-06432-t003:** Main characteristics of previous studies which are present in the literature.

Study	n° Pz	n° Proc.	Follow-Up (Months)	Embolizing Materials	Complications	Technical Success	Recurrences	Death	Conclusions
Cohen 1990 [6]	20	36	37	Coils, Gelfoam, PVA	Coil migration, chest pain, dysphagia, fever	19 (100%)	8	4	Safe and Effective
Sweezey 1990 [4]	25	38	/	Gelfoam, Coils	Mild fever, chest pain	21 (84%)	13	6	Safe and Effective
Tonkin 1991 [15]	11	20	9 to 108	Coils, Gelfoam, PVA	Dysphagia, chest pain, pseudoaneurysm femoral artery	11 (100%)	5	3	Safe and Effective
Cipolli 1995 [13]	14	17	10.5	PVA	Chest pain, fever, dysphagia	10 (71.4%)	3	3	Safe and Effective
Brinson 1998 [1]	18	36	22	Coils, Gelfoam, PVA	Chest pain, dysphagia	18 (100%)	3	2	Safe and Effective
Antonelli 2002 * [3]	8	/	36	PVA	Chest pain, fever	8 (100%)	/	1	Safe and Effective
Barben 2002 [2]	20	38	61	Coils, Gelfoam, PVA	/	36 (95%)	11	3	Safe and Effective
Vidal 2006 [5]	30	42	60	Microspheres, PVA	Chest pain	29 (96.6%)	8	1	Safe and Effective
Flight 2017 [10]	27	51	26	PVA	Chest pain, paresthesia, limb weakness	27 (100%)	10	9	Safe and Effective
Martin 2020 [14]	28	38	58	TAG Microspheres, PVA	Chest pain, odynophagia, femoral artery thrombosis	25 (89%)	7	/	Safe and Effective
Dohna 2021 [11]	34	70	89.9	Coils	Mild cough, chest pain	34 (100%)	20	12	Safe and Effective
Silveira 2021 [12]	17	39	130	/	Chest pain, fever, dyspnea	17 (100%)	8	3	Safe and Effective

*n° pz* number of patients, *n° proc* number of procedures, *PVA* polyvinyl alcohol particles. * This was the only study in the literature that evaluates the usage of BAE to treat sub-massive hemoptysis in CF patients.

## Data Availability

Not applicable.

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
