# Peer review of "Role of Bronchial Artery Embolization as Early Treatment Option in Stable Cystic Fibrosis Patients with Sub-Massive Hemoptysis: Personal Experience and Literature Review"

_jcm, 2022, doi:10.3390/jcm11216432_

Round 1

Reviewer 1 Report

This article investigates the role of bronchial artery embolization as early treatment option in stable cystic fibrosis patients with submassive hemoptysis. Although the topic is interesting, the context of the research is not sufficiently described and the discussion in several parts is a mere repetition of the results.

Specific comments:

- Introduction

The introduction is divided into too many paragraphs, paragraphs should be combined.

The MAV's full name must be written and MAV followed by parentheses.

- Materials and Methods

The FOV's full name must be written and FOV followed by parentheses.

- Results

In Table 3, "N° pz", "N° proc.", "PVA" and "TAG" explanations should be written below the table.

In Figure 4, differences between groups should be marked with *. Statistical significance level and method should be added to the subtitle of Figure 4.

- Discussion

Discussion section in the present form is a repetition of the results. If possible, this section should critically analyse and interpret the findings of the study.

Reviewer 2 Report

Dear Authors,

-It would be appropriate to indicate the number of patients included in the study and their gender distribution in the patient selection section. How many patients the study was conducted on is understood later in the manuscript. Ethics committee approval was obtained for this study.

-The acceptance number of the ethics committee should be indicated in parentheses.

-The English of the manuscript should be reviewed and typos corrected (eg line-154 salective-selective).

-N, indicating the number of cases, should be written in lower case (such as n=13-line 227, n=19-line 241).

-In the initial embolization period, respiratory infection would be more appropriate instead of ‘chronic respiratory infection’ terminology for infection or coinfection. Or did the patients really have a chronic lung infection?

-Do the abbreviations Y and N mean yes and no under the heading where the significance importance of right and left vessel calibers are indicated in the table? (Table 2). The explanations of the abbreviations in the table should be stated under or inside the table.
